



# Groundwater Level Forecasting with Artificial Neural Networks: A Comparison of LSTM, CNN and NARX

Andreas Wunsch[1], Tanja Liesch[1], and Stefan Broda[2]

[1]Karlsruhe Institute of Technology (KIT), Institute of Applied Geosciences, Division of Hydrogeology, Kaiserstr. 12, 76131 Karlsruhe, Germany
[2]Federal Institute for Geosciences and Natural Resources (BGR), Wilhelmstr. 25-30, 13593 Berlin, Germany

**Correspondence:** Andreas Wunsch (andreas.wunsch@kit.edu)

**Abstract.** It is now well established to use shallow artificial neural networks (ANN) to obtain accurate and reliable groundwater level forecasts, which are an important tool for sustainable groundwater management. However, we observe an increasing shift from conventional shallow ANNs to state-of-the-art deep learning (DL) techniques, but a direct comparison of the performance is often lacking. Although they have already clearly proven their suitability, especially shallow recurrent networks

frequently seem to be excluded from the study design despite the euphoria about new DL techniques and its successes in various disciplines. Therefore, we aim to provide an overview on the predictive ability in terms of groundwater levels of shallow conventional recurrent ANN namely nonlinear autoregressive networks with exogenous inputs (NARX), and popular state-of-the-art DL-techniques such as long short-term memory (LSTM) and convolutional neural networks (CNN). We compare both the performance on sequence-to-value (seq2val) and sequence-to-sequence (seq2seq) forecasting on a 4-year period, while

using only few, widely available and easy to measure meteorological input parameters, which makes our approach widely applicable. We observe that for seq2val forecasts NARX models on average perform best, however, CNNs are much faster and only slightly worse in terms of accuracy. For seq2seq forecasts, mostly NARX outperform both DL-models and even almost reach the speed of CNNs. However, NARX are the least robust against initialization effects, which nevertheless can be handled easily using ensemble forecasting. We showed that shallow neural networks, such as NARX, should not be neglected in

comparison to DL-techniques; however, LSTMs and CNNs might perform substantially better with a larger data set, where DL really can demonstrate its strengths, which is rarely available in the groundwater domain though.

## 1 Introduction

Groundwater is the only possibility for 2.5 billion people worldwide to cover their daily water needs (UNESCO, 2012) and at

least half of the global population uses groundwater for drinking water supplies (WWAP, 2015). Moreover, groundwater also constitutes for a substantial amount of global irrigation water (FAO, 2010), which altogether and among other factors such as population growth and climate change, make it a vital future challenge to dramatically improve the way of using, managing





and sharing water (WWAP, 2015). Accurate and reliable groundwater level (GWL) forecasts are a key tool in this context, as
they provide important information on the quantitative availability of groundwater and can thus form the basis for management
decisions and strategies.

Especially due to the success of deep learning (DL) approaches in recent years and their more and more widespread ap-
plication in our daily life, DL starts to transform traditional industries and is also increasingly used across multiple scientific
disciplines (Shen, 2018). This applies as well to water sciences, where machine learning methods in general are used in a
variety of ways, as data-driven approaches offer the possibility to directly address questions on relationships between relevant
input forcings and important system variables, such as run-off or groundwater level, without the need to build classical models
and explicitly define physical relationships. This is especially handy because these classical models sometimes might be over-
simplified or in the case of numerical models, data-hungry, difficult and time-consuming to set up and maintain, and therefore
expensive. In particular artificial neural networks (ANNs) have been successfully applied to a variety of surface water (Maier
et al., 2010) and groundwater level (Rajaee et al., 2019) related research questions already; however, especially DL was used
only gradually at first (Shen, 2018), but is just about to take off, which is reflected in the constantly increasing number of DL
and water resource-related publications. In this work we explore and compare the abilities of nonlinear autoregressive models
with exogenous input (NARX), which have been successfully applied multiple times to groundwater level forecasting in the
past, to the currently popular DL approaches long short-term memory (LSTM) and convolutional neural networks (CNN).
During the last years several authors have shown the ability of NARX to successfully model and forecast groundwater levels
(Alsumaiei, 2020; Chang et al., 2016; Di Nunno and Granata, 2020; Guzman et al., 2017, 2019; Hasda et al., 2020; Izady et al.,
2013; Jeihouni et al., 2019; Jeong and Park, 2019; Wunsch et al., 2018; Zhang et al., 2019). Although LSTMs and CNNs are
state-of-the-art DL techniques and commonly applied in many disciplines, they are not yet widely adopted in groundwater level
prediction applications, if so, but within the last two years. Thereby, LSTMs were used twice as often to predict groundwater
levels (Afzaal et al., 2020; Bowes et al., 2019; Jeong et al., 2020; Jeong and Park, 2019; Müller et al., 2020; Supreetha et al.,
2020; Zhang et al., 2018) compared to CNNs (Afzaal et al., 2020; Lähivaara et al., 2019; Müller et al., 2020). The main reason
might be that the strength of CNNs is mainly the extraction of spatial information from image like data, whereas LSTMs are
especially suited to process sequential data, such as from time-series. Overall, these studies show that as well LSTMs as CNNs
are very well suited to forecast groundwater levels. Both, Afzaal et al. (2020) and Müller et al. (2020) also directly compared
the performance of LSTMs and CNNs but no clear superiority of one to the other can be drawn from their results. In the case
of Afzaal et al. (2020) the results are very similar and extensive interpretations are hampered because of the comparably hard
presentation style of the results (no easy interpretable metrics such as Nash-Sutcliffe efficiency, and hard to read plotting style).
Müller et al. (2020), who focus on hyperparameter optimization, draw the conclusion that CNN results are less robust compared
to LSTM predictions, however, other analyses in their study also show better results of CNNs compared to LSTMs. Jeong and
Park (2019) conducted a comparison of NARX and LSTM (and others) performance on groundwater level forecasting. They
found both to be the best models in their overall comparison concerning the prediction accuracy, however, they used a deep
NARX model with more than one hidden layer. To the best of the authors' knowledge, no direct comparison has yet been made
of (shallow) NARX, LSTMs and CNNs to predict groundwater levels.





In this study we aim to provide an overview on the predictive ability in terms of groundwater levels of shallow conventional recurrent ANN namely NARX, and popular state-of-the-art DL-techniques LSTM and (1D-) CNN. We compare both
the performance on single value (sequence-to-value) and sequence (sequence-to-sequence) forecasting. We use data from 17 groundwater wells within the Upper Rhine Graben region in Germany and France, selected based on prior knowledge and representing the full bandwidth of groundwater dynamic types in the region. Further we use only widely available and easy to measure meteorological input parameters (precipitation, temperature and relative humidity), which makes our approach widely applicable.

## 2   Methodology

### 2.1   Input Parameters

In this study we only use the meteorological input variables precipitation (P), temperature (T) and relative humidity (rH), which in general are widely available and easy to measure. In principle, this makes this approach easily transferable and thus applicable almost everywhere. Precipitation may serve as a surrogate for groundwater recharge, temperature and relative humidity
include the relationship with evapotranspiration and at the same time provide the network with information on seasonality due to the usually distinct annual cycle. As an additional synthetic input parameter, a sinusoidal signal fitted to the temperature curve (Tsin), can provide the model with noise-free information on seasonality, which often allows significantly improved predictions to be made (Kong A Siou et al., 2012). Without doubt, the most important input parameter out of these is P, since groundwater recharge usually has the greatest influence on groundwater dynamics. Therefore, P is used always as input pa-
rameter, the suitability of the remaining parameters is checked and optimized for each time series and each model individually. The fundamental idea is that for wells with primarily natural groundwater dynamics, the relationship between groundwater levels and the important processes of groundwater recharge and evapotranspiration should be mapped via the meteorological parameters P, T and rH. However, especially for wells with a dynamic influenced by other factors, this is usually only valid to a limited extent, since groundwater dynamics can depend on various additional factors such as groundwater extractions or
surface water interactions. Due to a typically strong autocorrelation of groundwater level time series, a powerful predictor for the future groundwater level is the groundwater level in the past. Depending on the purpose and methodological setup it does not always make sense to include this parameter; however, where meaningful we explored also past GWL as Inputs.

### 2.2   Nonlinear autoregressive exogenous model (NARX)

Nonlinear autoregressive networks with exogenous input are a specific type of recurrent neural networks (RNNs) that extend the
well-known structure of feed-forward multilayer perceptrons (MLP) by a global feedback connection between output and input layer. They are frequently applied for nonlinear time series prediction and nonlinear filtering tasks (Beale et al., 2016). Similar to other types of RNNs, NARX have also difficulties in capturing long-term dependencies due to the problem of vanishing and exploding gradients (Bengio et al., 1994), yet they can keep information up to three times longer than simple RNNs (Lin





et al., 1996a, b), so they can converge more quickly and generalize better in comparison (Lin et al., 1998). Using the recurrent

connection, future outputs are both regressed on independent inputs and on previous outputs (groundwater levels in our case), which is the standard configuration for multi-step prediction and also known as closed-loop configuration. However, NARX can also be trained by using the open-loop configuration, where the observed target is presented as an input, instead of feeding back the estimated output. This configuration can make training more accurate and efficient, as well as computationally less expensive, because learning algorithms do not have to handle recurrent connections (Moghaddamnia et al., 2009). However,

experience shows that both configurations can be adequate for training a NARX model, since open-loop training often results in more accurate performance in terms of mean errors, whereas closed-loop trained models often are better in capturing the general dynamics of a time-series. NARX also contain a short-term memory, i.e. delay vectors for each input (and feedback), which allow the availability of several input time steps simultaneously, depending on the length of the vector. Usually, delays are crucial for the performance of NARX models.

The given configuration describes sequence-to-value forecasting, to perform sequence-to-sequence forecasts, some modifications are necessary. As other ANN, NARX are capable of perform forecasts of a complete sequence at once, i.e. one output neuron predicts a vector with multiple values. Technically it is necessary to use the same length for input and output sequences. To build and apply NARX models, we use Matlab 2020a (Mathworks Inc., 2020) and its Deep Learning Toolbox.

## 2.3 Long Short-Term Memory (LSTM)

Long Short-Term Memory networks are recurrent neural networks which are widely applied to model sequential data like time series or natural language. As stated, RNNs suffer from the vanishing gradient problem during backpropagation and in the case of simple RNNs, their memory barely includes the previous 10 time-steps (Bengio et al., 1994). LSTMs, however, can remember long-term dependencies because they have been explicitly designed to overcome this problem (Hochreiter and Schmidhuber, 1997). Besides the hidden state of RNNs, LSTMs have a cell memory (or cell state) to store information and

three gates to control the information flow (Hochreiter and Schmidhuber, 1997). The forget gate (Gers et al., 2000) controls which and how much information of the cell memory is forgotten, the input gate controls which inputs are used to update the cell memory, and the output gate controls which elements of the cell memory are used to update the hidden state of the LSTM cell. The cell memory enables the LSTM to handle long-term dependencies because information can remain in the memory for many steps (Hochreiter and Schmidhuber, 1997). Several LSTM layers can be stacked on top of each other in a model, however

the last LSTM layer is followed by a traditional fully connected dense layer, which in our case is a single output neuron that outputs the groundwater level. To realize sequence forecasting, as many output neurons in the last dense layer as steps in the sequence are needed. For LSTMs we rely on Python 3.8 (van Rossum, 1995) in combination with the libraries Numpy (van der Walt et al., 2011), Pandas (McKinney, 2010; Reback et al., 2020), Scikit-Learn (Pedregosa et al., 2011) and Matplotlib (Hunter, 2007). Further we use the Deep-Learning frameworks TensorFlow (Abadi et al., 2015) and Keras (Chollet, 2015).





## 2.4 Convolutional Neural Networks (CNN)

CNNs are neural networks, which are predominantly used for image recognition and classification. However, they work also
well on signal processing tasks and are used for Natural Language Processing for example. CNNs usually comprise three
different layers. Convolutional layers, the first type, consist of filters and feature maps. The input to a filter is called receptive
field and has a fixed size. Each filter is dragged over the entire previous layer resulting in an output, which is collected
in the feature map. Convolutional layers are often followed by pooling layers that perform down-sampling of the previous
layers feature map, thus information is consolidated by moving a receptive field over the feature map. Such fields apply simple
operations like averaging or maximum selection. Similar to LSTM models, multiple convolutional and pooling layers in varying
order can be stacked on top of each other in deeper models. The last layer is followed by a fully connected dense layer with
one or several output neurons. To realize sequence forecasting, as many output neurons in the last dense layer as steps in the
sequence are needed. For CNNs we equally use Python 3.8 (van Rossum, 1995) in combination with the above mentioned
libraries and frameworks.

## 2.5 Model Calibration and Evaluation

In this study we use NARX models with one hidden layer, and train them in closed loop using the Levenberg-Marquardt
algorithm, which is a fast and reliable second-order local method (Adamowski and Chan, 2011). We choose close-loop con-
figuration for training, because other hyperparameters (HPs) are optimized using a Bayesian model (see below), which seems
to work properly only in closed-loop configuration, probably due to the artificially pushed training performance in open-loop
configuration. Optimized HPs are the inputs T, Tsin and rH (1/0, i.e. yes/no), size of the input delays (ID P, ID T, ID Tsin,
ID rH), size of the feedback delay vector (FD) and number of hidden neurons (hidden size). Delays (ID/FD) can take values
between 1 and 52 (which is one year of weekly data), the number of hidden neurons is optimized between 1 and 20. Strictly
speaking, input selection is no hyperparameter optimization problem, however, the algorithm can also be applied to select an
appropriate set of inputs. This assumption applies in our study also to LSTM and CNN models.

We choose our LSTM models to consist of one LSTM layer, followed by a fully connected dense layer with a single output
neuron in the case of sequence-to-value forecasting. We use Adam-Optimizer with an initial learning rate of 1E-3 and apply
gradient clipping to prevent gradients from exploding. Hyperparameters being optimized by a Bayesian model are the number
of units within the LSTM layer (hidden size, 1 to 256), the batch size (1 to 256) and the sequence length (1 to 52). The latter
can be interpreted more or less as equivalent to the delay size of the NARX models and is often referred to as the number of
inputs.

The CNN models we apply consist of one convolutional layer, a max-pooling layer, and two dense layers, where the second
one consists only of one neuron in case of sequence-to-value forecasting. Adam-optimizer is used with the same configuration
as for the LSTM models. For all CNN models we use a kernel size of 3 and optimize the batch size (1 to 256), sequence length
(1 to 52), the number of filters (1 to 256) within the convolutional layer, as well as the number of neurons within the first dense
layer (dense size, 1 to 256) according to a Bayesian optimization model.





Hyperparameter Optimization is conducted by applying Bayesian optimization using the python implementation by Nogueira (2014). We apply 50 optimization steps as a minimum (25 random exploration steps followed by 25 Bayesian optimization steps). After that, the optimization stops as soon as no improvement has been recorded during 20 steps or after a maximum of 150 steps. For the NARX models we use the Matlab built-in Bayesian-optimization, where the first 50 steps cannot be distinguished as explained, however, the rest applies accordingly. The acquisition function in all three cases is expected improvement and the optimization target function we chose is the sum of Nash-Sutcliffe Efficiency (NSE) and coefficient of determination ($R^2$) (comp. equations 1 and 2), because these two criteria are very important and well established criteria for assessing the forecast accuracy in water-related contexts.

All three model types use 30 as the maximum number of training epochs. To prevent overfitting, we apply early stopping with a patience of 5 steps. The testing or evaluation period in this study for all models are the years 2012 to 2015 inclusively. This period is exclusively used for testing the models. The data before 2012 is of varying length (hydrographs start between 1967 and 1994), depending on the available data and is split into three parts, namely 80% for training, and as well 10% for early stopping as 10% for testing during HP-Optimization (opt-set) (Fig. 1). Thus, the target function of the HP-optimization procedure is only calculated on the opt-set.

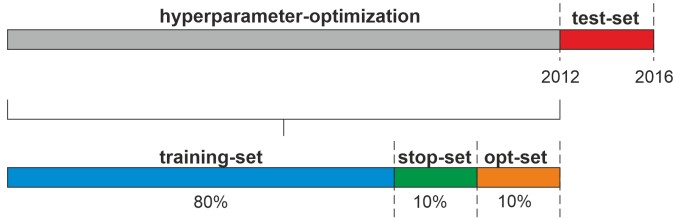

**Figure 1.** data splitting scheme

All data is scaled between -1 and 1 and all models are initialized randomly and show therefore a dependency towards the random number generator seed. To minimize initialization influence we repeat every optimization step 5 times and take the mean of the target function. For the final model evaluation in the test period (2012–2016) we use 10 pseudo-random initializations and calculate errors of the median forecast. For sequence2sequence forecasting we always take the median performance over all forecasted sequences, which have a length of 3 months or 12 steps respectively. This is a realistic length for direct sequence forecasting of groundwater levels, which also has some relevance in practice, because it (i) provides useful information for many decision-making applications (e.g. groundwater management), and (ii) is also an established time-span in meteorological forecasting, known as seasonal forecasts. In principle, this also allows a performance comparison of 12-step seq2seq forecasts with a potential 12-step seq2val forecast, based on operational meteorological forecasting, where the input uncertainty potentially lowers the groundwater level forecast performance. However, this is beyond the scope of this study, which focuses on neural network architecture comparison.

To judge forecast accuracy, we calculate several metrics: Nash-Sutcliffe Efficiency (NSE), coefficient of determination ($R^2$), absolute and relative root mean squared error (RMSE/rRMSE), absolute and relative Bias (Bias/rBias) as well as Persistency





180   index (PI). For the following equations applies that $o$ represents observed values, as well as $p$ represents predicted values, n stands for the number of samples.

$$NSE = 1 - \frac{\sum_{i=1}^{n} (o_i - p_i)^2}{\sum_{i=1}^{n} (o_i - \bar{o})^2} \tag{1}$$

  Please note that in the denominator we use the mean observed values until the start of the test-period (2012 in the case of our final model evaluation). This represents best the meaning of the NSE, which compares the model performance to the mean

185   values of all known values at the time of the start of the forecast.

$$R^2 = \left( \frac{\sum_{i=1}^{n} (o_i - \bar{o})(p_i - \bar{p})}{\sqrt{\sum_{i=1}^{n} (o_i - \bar{o})^2} \sqrt{\sum_{i=1}^{n} (p_i - \bar{p})^2}} \right)^2 \tag{2}$$

$$RMSE = \sqrt{\frac{1}{n} \sum_{i=1}^{n} (o_i - p_i)^2} \tag{3}$$

$$rRMSE = \sqrt{\frac{1}{n} \sum_{i=1}^{n} \left( \frac{o_i - p_i}{o_{max} - o_{min}} \right)^2} \tag{4}$$

190   $$Bias = \frac{1}{n} \sum_{i=1}^{n} (o_i - p_i) \tag{5}$$

$$rBias = \frac{1}{n} \sum_{i=1}^{n} \left( \frac{o_i - p_i}{o_{max} - o_{min}} \right) \tag{6}$$

$$PI = 1 - \frac{\sum_{i=1}^{n} (o_i - p_i)^2}{\sum_{i=1}^{n} (o_i - o_{last})^2} \tag{7}$$

  Please note that RMSE and Bias are useful to compare performances for a specific time series among different models, however only rRMSE and rBias are meaningful to compare model performance between different time series. The persistency

195   index PI basically compares the performance to a naïve model that uses the last known observed groundwater level at the time the prediction starts. This is particularly important to judge the performance, when past groundwater levels ($\text{GWL}_{(t-1)}$) are used as inputs, because especially in this case the model should outperform a naïve forecast (PI > 0).

## 2.6   Computational Aspects

We used different computational setups to build and apply the three model types. We built the NARX models in Matlab and

200   performed the calculations on the CPU (AMD-Ryzen 9 3900X). The use of a GPU instead of a CPU is not possible for NARX





models in our case, because of the Levenberg-Marquardt training algorithm, which is not suitable for GPU computation. Both LSTMs and CNNs, however, can be calculated on a GPU, which in the case of LSTMs is the preferred option. For CNNs we observed a substantially faster calculation (factor 2 to 3) on the CPU and therefore favoured this option. Both LSTMs and CNNs were built and applied using Python 3.8, the GPU we used for LSTMs was a Nvidia GeForce RTX 2070 Super.

## 3    Data and Study Area

In this study we examine the groundwater level forecasting performance at 17 groundwater wells within the Upper Rhine Graben (URG) area, the largest groundwater resource in central Europe (LUBW, 2006). The aquifers of the URG cover 80% of the drinking water demand of the region as well as the demand for agricultural irrigation and industrial purposes (Région Alsace - Strasbourg, 1999). The wells are selected from a larger dataset from the region with more than 1800 hydrographs. Based on prior knowledge, the wells of this study represent the full bandwidth of groundwater dynamics occurring in the dataset. The whole dataset mainly consists of shallow wells from the uppermost aquifer within the Quaternary sand/gravel sediments of the URG. Mean GWL depths are smaller than 5 m bgl. for 70% of the data, rising to a maximum of about 20-30 m towards the Graben edges. The considered aquifers show generally high storage coefficients and high hydraulic conductivities in the order of 10E-4 to 10E-3 m/s (LUBW, 2006). In some areas, e.g. the northern URG, strong anthropogenic influences exist, due to intensive groundwater abstractions and management efforts. A list of all examined wells with additional information on identifiers and coordinates can be found in the supplement (Table S1). All groundwater data is available for free via the web services of the local authorities (HLNUG, 2019; LUBW, 2018; MUEEF, 2018). The shortest time series starts in 1994, the longest in 1967, however, most hydrographs (12) start between 1980 and 1983. Meteorological input data was derived from the HYRAS dataset (Frick et al., 2014; Rauthe et al., 2013), which can be obtained free of charge for non-commercial purposes on request from the German Meteorological Service (DWD). In this study we exclusively consider weekly time steps for both groundwater and meteorological data.

## 4    Results and Discussion

### 4.1    Sequence-to-value (seq2val) forecasting performance

Figure 2 summarizes and compares the overall seq2val forecasting accuracy of the three model types for all 17 wells. Fig. 2a shows the performance when only meteorological inputs are used, the models in Fig. 2b are additionally provided with $GWL_{t-1}$ as an input. Because the GWL of the last step has to be known, the latter configuration has only limited value for most applications since only one-step-ahead forecasts are possible in a real-world scenario. However, the inputs of the former configuration are usually available as forecasts themselves for different time horizons. Fig. 2a shows that on average NARX models perform best, followed by CNN models, LSTMs achieve the least accurate results. This is consistent for all error measures except rBias, where CNN models show slightly less bias than NARX. However, all models suffer from significant negative bias values in the same order of magnitude, meaning that GWLs are systematically underestimated. Providing information about past





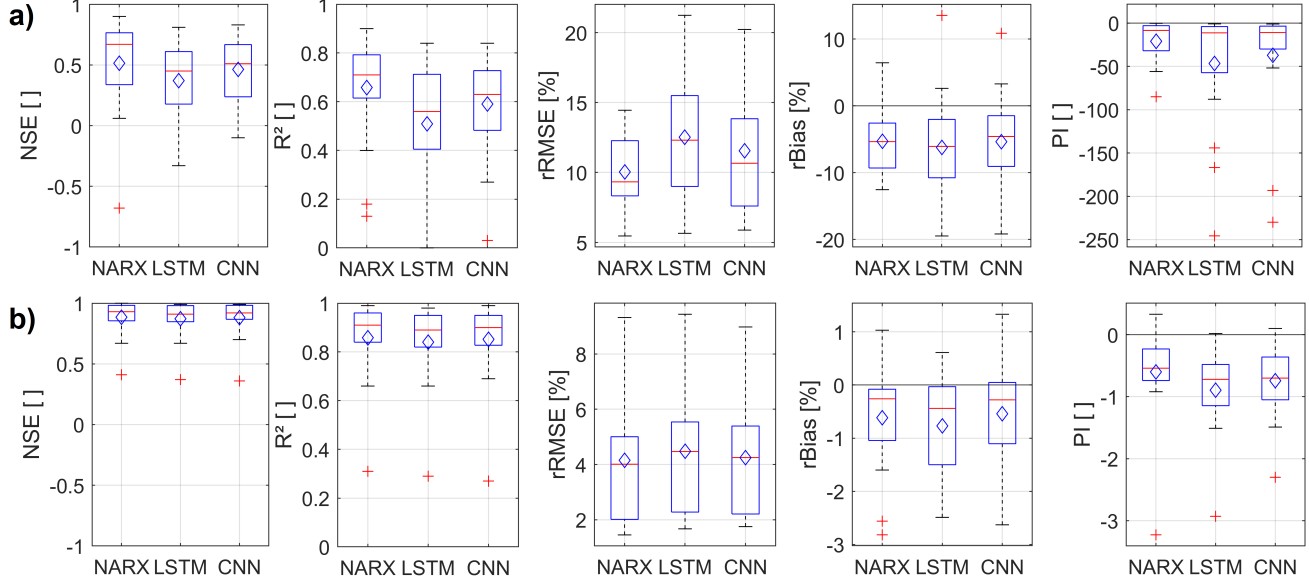

**Figure 2.** Boxplots showing the seq2val forecast accuracy of NARX, LSTM and CNN models within the test period (2012-2016) for all considered 17 hydrographs. The diamond indicates the arithmetic mean; a) only meteorological inputs; b) $GWL_{t-1}$ as additional Input

groundwater levels up to t-1 ($GWL_{t-1}$), improves the performance of all three models significantly (Fig. 2b). Additionally, performance differences between the models vanish and remain only visible as slight tendencies. This is not surprising, as the past groundwater level is usually a good or even the best predictor of the future GWL, at least for one step ahead and all

models are able to use this information. The general superiority of NARX in case of Fig. 2a is therefore also not surprising, as a feedback connection within the model already provides information of past groundwater levels, even though it includes also a certain forecasting error. However, providing $GWL_{t-1}$ as input to a seq2val-model (Fig. 2b) basically means providing the naïve model itself, which needs to be outperformed in the case of PI metric (comp. Sec. 2.5). PI values below zero therefore basically mean that the output is worse than the input, which is, apart from the limited benefit for real applications mentioned

above, why we refrain from further discussion of the models shown in Fig. 2b.

     For our analysis we did not make a pre-selection of hydrographs that show predominantly natural groundwater dynamics and thus a comparatively strong relationship between the available input data and the groundwater level. Therefore, even though hydrographs possibly influenced by additional factors were examined, we can conclude that the forecasting approach in general works quite well and we reach e.g. median NSE values of $\geq 0.5$ for NARX and CNNs, LSTMs show a median value

only slightly lower. In terms of robustness against the initialisation dependency of all models (ensemble variability), we clearly observe the highest dependency for NARX, followed by CNN and LSTM, while LSTMs on median perform slightly more robust than CNNs. Including $GWL_{t-1}$ lowers the error variance of the ensemble members, which we used to judge robustness in this case, by several orders of magnitude for all models. NARX and LSTMs on median now show slightly lower ensemble





variability than CNNs, however all models are quite close. A corresponding figure was added to the supplement (Figure S69).
Furthermore, we added information on the results of the hyperparameter-optimization (tables S2-S4), a table with all error measure values of each considered hydrograph and model (table S5), as well as according seq2val forecasting plots (Figures S1 to S34) to the supplement, as well.

Figure 3 shows exemplarily the forecasting performance of all three models for well BW_104-114-5, where all three models consistently achieved good results in terms of accuracy. The NARX model (a) outperforms both LSTM (b) and CNN (c) models
and shows very high NSE and $R^2$ values between 0.8 and 0.9. The CNN model provides the second best forecast, which even very slightly shows less underestimation (Bias/rBias) of the GWLs than the NARX model. By comparing the graphs in (a) and (c) we assume that this is only true on average. The CNN model overestimates in 2012 and constantly underestimates the last third of the test period. The NARX model, however, is more consistent and therefore better. Concerning $R^2$ values, the LSTM basically keeps up with the CNN, all other error measures show the still good, but in comparison worst values. We notice that
in accordance to our overall findings mentioned above, the LSTM shows the lowest ensemble variability and therefore the smallest initialization dependency. Taking a look at the selected inputs and hyperparameters, we notice that rH does not seem to provide important information and was therefore never selected as an input. Further the input sequence length of both LSTM and CNN is equally 35 steps (weeks). In the NARX model there is no direct correspondence, but a similar value is shown by the parameter FD and thus the number of past predicted GWL values available via the feedback connection.

In contrast to the above-mentioned well, hardly any systematic can be derived from the choice of input parameters across all wells that even might have physical implications for each site. Rather, it is noticeable that certain model types seem to prefer also certain inputs. For example, temperature is only selected as input in 5 out of 17 cases for LSTM models, and in 2 out of 17 cases for CNN models. Furthermore, relative humidity (rH) is always selected for LSTM models except for two times. In case of NARX models there seems to be a lack of systematic behaviour. For more details please see tables S2-S4 in the supporting
material.

Our approach assumes a groundwater dynamic mainly dominated by meteorological factors. We can assume that all three model types are basically capable of modelling groundwater levels very accurately if all relevant input data can be identified. To exemplarily show the influence of additional input variables, which, however, are usually not available as input for a forecast or even have insufficient historical data, Fig. 4 illustrates the significantly improved performance after including the Rhine River
water level (W), a large streamflow within the study area, using the example of the NARX model for well BW_710-256-3, which indeed is located close to the river. Besides improved performance, we also observe lower variability of the ensemble member results, thus, lower dependency to the model initialization, which corresponds also to other time-series, where we often find smaller influence the more relevant the input data is. This also confirms that little accuracy is probably due to insufficient input data on a case-by-case basis, not necessarily because of an inadequate modelling approach.





**Figure 3.** Forecasts of (a) a NARX, (b) a LSTM and (c) a CNN model for well BW_104-114-5 during the test period 2012-2016.





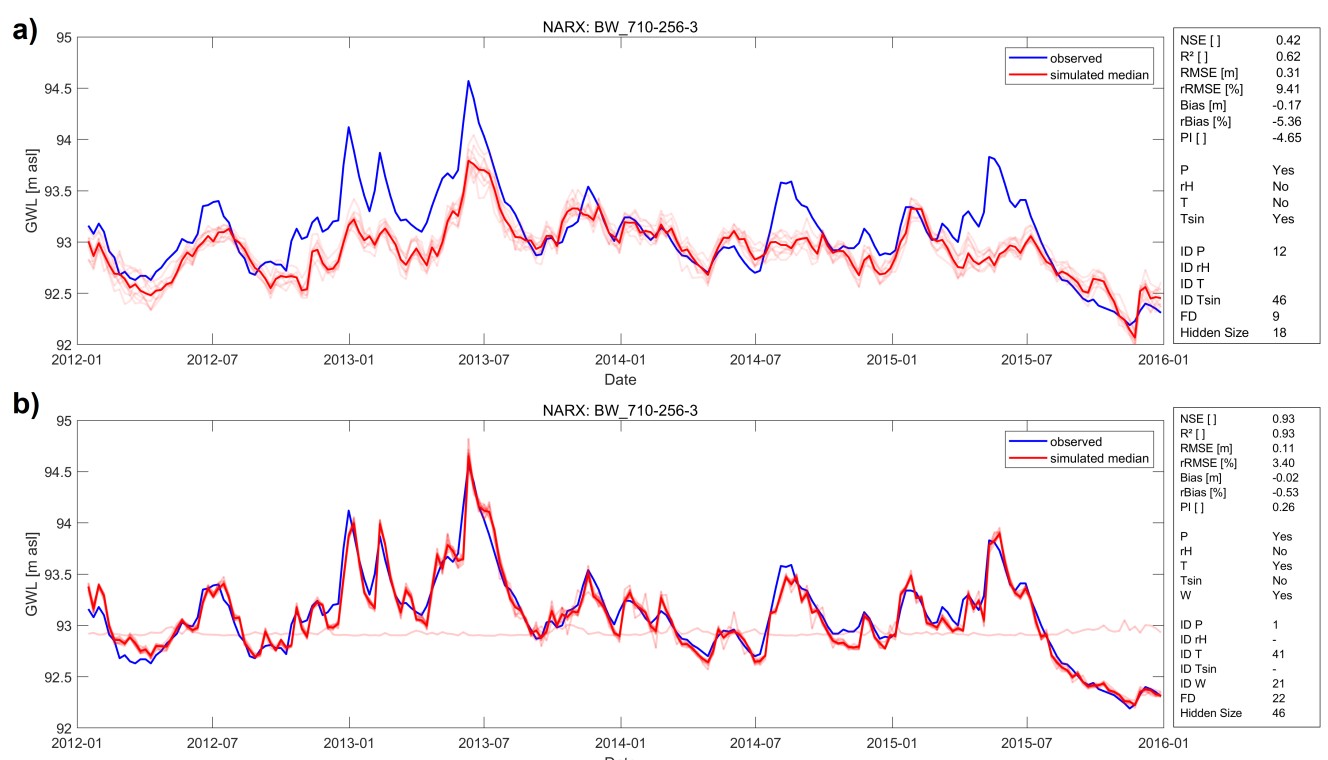

**Figure 4.** Forecasting performance exemplarily shown for NARX model of well BW_710-256-3 (a) based on meteorological input variables and (b) improved performance after including Rhine River water level (W) as input variable.





## 4.2 Sequence-to-sequence (seq2seq) forecasting performance

Sequence-to-sequence forecasting is especially interesting for short- and mid-term forecasts because the input variables only have to be available until the start of the forecast. Figure 5 summarizes and compares the overall seq2seq forecasting accuracy of the three model types for all 17 wells. Fig. 5a shows the performance when only meteorological inputs are used, the models in Fig. 5b are additionally provided with $GWL_{t-1}$ as an input. Equally to the seq2val forecasts (Fig. 2) past GWLs seem to be especially important for LSTM and CNN models, where this additional input variable causes substantial performance improvement. Without past GWLs, NARX seem to be clearly superior due to their inherent global feedback connection. However, NARX show almost equal performance values in both scenarios (Fig. 5a and b). In contrary to the seq2val forecasts, for seq2seq forecasts NARX systematically show lower $R^2$ values than LSTM and CNN models. For all other error measures, the accuracy of NARX models outperforms LSTMs and CNNs in a direct comparison for the vast majority of all time series. While LSTMs and CNNs show lower performance for sequence-to-sequence forecasting compared to sequence-to-value forecasting, NARX seq2seq models even outperform NARX seq2val models (except for $R^2$). This is quite counter-intuitive as one would expect it to be more difficult to forecast a whole sequence than a single value. All in all, the scenario including past GWLs (Fig. 5b) seems to be the preferable one for all three models and shows promising results for real world applications. Detailed results on all seq2seq models can be found in the supplementary material (Table S6, Figures S35 to S68).

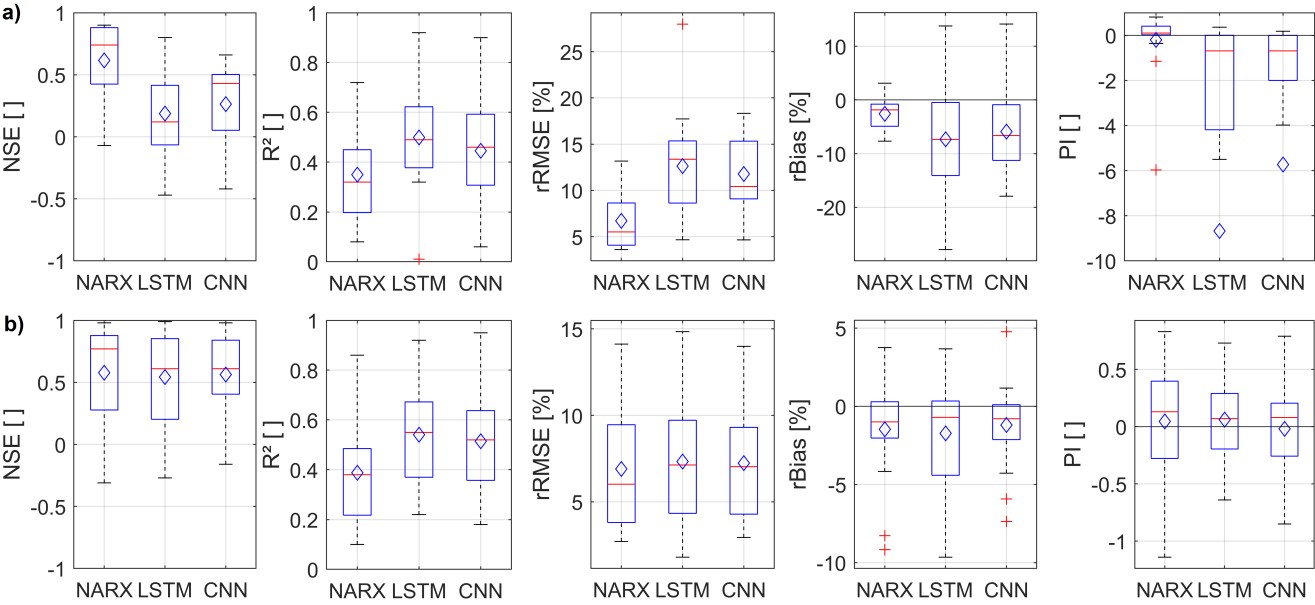

**Figure 5.** Boxplots showing the seq2seq forecast accuracy of NARX, LSTM and CNN models within the test period (2012-2016) for all considered 17 hydrographs. The diamond indicates the arithmetic mean; a) only meteorological inputs; b) $GWL_{t-1}$ as additional Input.





Figure 6 summarises exemplarily for well HE_11874 the sequence-to-sequence forecasting performance for NARX (a,b), LSTMs (c,d), CNNs (e,f), only with meteorological input variables (a,c,e) and with an additional past GWL input (b,d,f). These confirm, that $GWL_{t-1}$ substantially improves the performance of LSTMs and CNNs, however, NARX forecasts in this case only improve very slightly. Especially for LSTMs and CNNs it is easily visible that the sequence forecasts of the better models (d,f) mostly estimate the intensity of a future groundwater level change too conservatively, thus both in- and decreases are predicted too weak. This is a commonly known issue with ANN, as extreme values are typically under-represented in the

distribution of the training data (e.g Sudheer et al., 2003). We further notice that the robustness of LSTMs and CNNs in terms of initialization dependency, thus the ensemble variability, significantly improves when past GWLs are provided as inputs (Fig. 6). This is also supported by analysing the ensemble-member error variances and also true for all other time series in the dataset as well (Figure S69 in the supplement). Just like for seq2val forecasts, NARX usually show a significantly lower robustness in

terms of initialization dependency; however, the median ensemble performance nevertheless is of high accuracy. All models, but especially NARX models, therefore should not be evaluated without including an initialization ensemble. The initialization dependency of LSTMs and CNNs is significantly lower, with LSTMs being even more robust than CNNs.

  The extraordinary performance of the NARX models, especially in case of Well HE_11874 (Fig. 6) surprises, because the performance substantially outperforms the seq2val NARX without $GWL_{t-1}$ input (e.g. NSE: 0.35, $R^2$: 0.75); however, the

seq2val NARX model with $GWL_{t-1}$ inputs also showed high accuracy (e.g. NSE: 0.99, $R^2$: 0.99). It is also interesting to note that the sequence predictions of the NARX models overlap exactly and the individual sequences are therefore no longer visible. One reason for this different behaviour compared to the LSTM and CNN models is probably that the technical approach for seq2seq forecasting differs for these models. While LSTMs and CNNs use multiple output neurons to predict multiple steps at once, this approach for us did not yield meaningful results for a NARX model, probably because of feedback connection

issues. Instead we used one NARX output neuron to predict a multi-element vector at once.

### 4.3 Hyperparameter Optimization and Computational Aspects

During the HP-Optimization, depending on the forecasting approach (seq2val/seq2seq) and available inputs (with/without $GWL_{t-1}$), there were noticeable differences with regard to the number of iterations required and the associated time needed (Fig. 7). The best parameter combination, especially for CNN and LSTM networks, was often found in 33 steps or fewer,

hence after 25 obligatory random exploration steps in only 8 Bayesian steps. Please note that prior to the analysis we chose to at least perform 50 optimization steps, which explains the distribution in the 'total iterations' column. In column two ('best iteration') we can observe similar behaviour of CNNs and LSTMs, NARX are always somehow different to these two. We suspect that this is rather an influence of the software or the optimization algorithm, since especially model types implemented in Python show an identical behaviour. However, in the majority of cases the best iteration was found in less than 33 steps, the

minimum as well as the maximum number of iteration steps were therefore obviously sufficient. It is interesting that for CNNs and LSTM the number of steps is similar throughout the experiments, whereas for NARX the inclusion of $GWL_{t-1}$ as input caused an increase of iterations. Columns three to five in Fig. 7 show substantial differences concerning the calculation speed of the three model types. CNNs outperform all other models systematically, however, concerning the sequence-2-sequence

**Figure 6.** Forecasts of (a,b) a NARX, (c,d) a LSTM and (e,f) a CNN model for well BW_104-114-5 during the test period 2012-2016; Models in (a,c,e) use only meteorological input variables, models (b,d,f) use also past GWL observations



forecasts, NARX models can almost keep up. We also observe that LSTMs seem to slow down when including $GWL_{t-1}$ as
input or when performing seq2seq forecasts, the opposite happens in case of NARX models, which speed up in these cases.
This also means, that even though NARX models need more optimization iterations until the assumed optimum than LSTMs,
in terms of time they outperform them due to shorter duration per iteration (col. 3). Please note that it is out of the scope of
this work to provide detailed assessments of the calculation speed under bench mark conditions, but to share practice relevant
insights for fellow hydrogeologists.

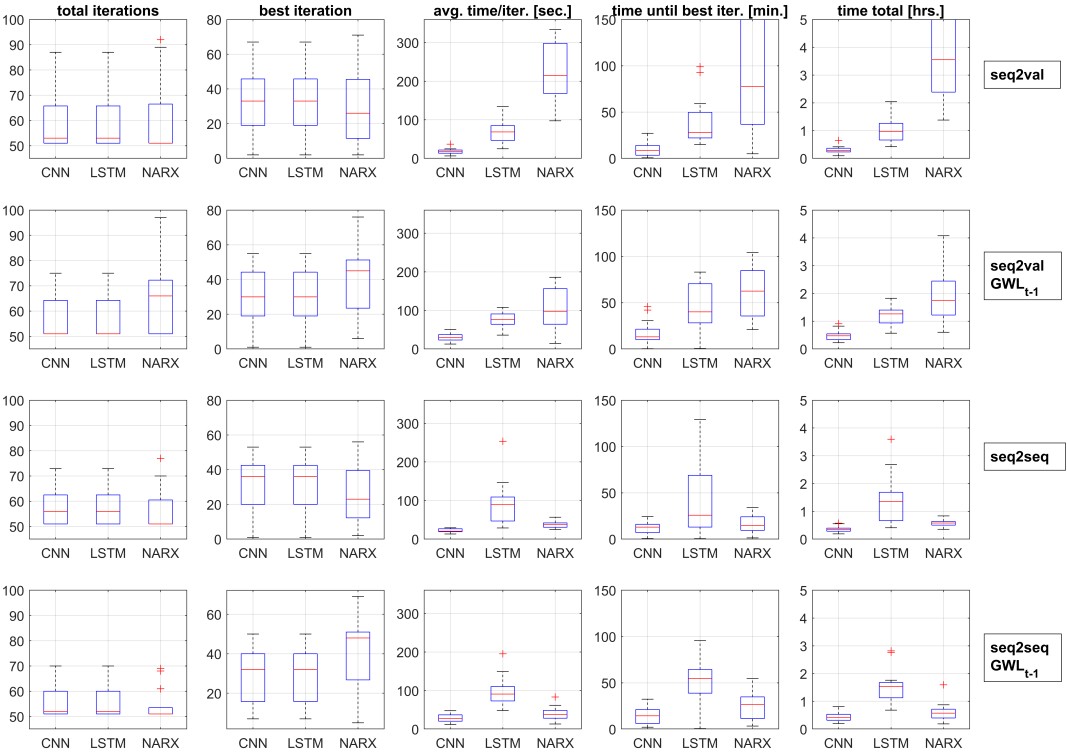

**Figure 7.** Comparison of the performed HP-optimizations (columns 1 and 2), their calculation time per iteration in seconds (col 3), until the
optimum was found (minutes) (col. 4) and the total time spent on optimization in hours (col. 5).

## 4.4  Influence of Training Data Length


In the following section we explore similarities and differences of NARX, LSTMs and CNNs in terms of the influence of
training data length. It is commonly known that data driven approaches profit from additional data, however, it still remains an
open question how much data is necessary to build models, which are able to perform reasonable calculations. This is, because
the answer is highly dependent on the application case, data properties (distribution e.g.) and model properties, as model-depth
can sometimes exponentially decrease the need for training data (Goodfellow et al., 2016). Therefore this question cannot be
entirely answered by a simple analysis, like we perform it here. Nevertheless, we still want to give an impression on how much





data might be approximately needed in the case of groundwater level data in porous aquifers and if the models substantially differ in their need for training data. For our analysis we always consider the forecasting accuracy during the 4-year testing period (2012-2016) and systematically expand the training data basis year by year, starting in 2010, thus with only clearly

insufficient two years of training data. We focus on sequence-to-value forecasting due to the easier interpretability of the results and we always consider the median performance of 10 different model initializations for evaluation. Figure 8 summarizes the performance and the improvement that comes with additional training data, all values are normalized per well to make them comparable. Please note that all models at least show 28 years of training data (until 1982), only 3 models exceed 30 years of training data (1980), thus the number of samples represented by the boxplots decreases significantly after 30 years. Fig. 8

summarizes as well models with as without $GWL_{t-1}$ inputs because no significantly different behaviour was observed for each group. Please find according figures for each group in the supplement (Figures S70-S71).

As expected, we observe significant improvements with additional training data. NARX models seem to improve more or less continuously, whereas for LSTMs and CNNs some kind of threshold is visible (about 10 years, thus approx. 500 samples), where the performance significantly increases and rapidly approaches the optimum. It should be noted though, that this can

probably not be transferred to other time steps, i.e. in the case of daily values e.g., 500 days will most certainly not be enough, since only one full yearly cycle is included. We explored the reason for this threshold and observed that when stopping the training five years earlier (2007), the threshold now occurs correspondingly five years earlier (figure S72 in supplement). Additionally, we found that several standard statistic values such as mean, median, variance, overall maximum, and the 25th, 75th as well as the 97.5th quantile show similar thresholds (figure S73 in supplement). Thus, the early years of the 2000s, seem

to be especially relevant for our test period. This is a highly dataset-specific observation that cannot be generalized; however, this also shows that it is vital to include relevant training data, which is, however, not very easy to identify. Nevertheless as a rule of thumb the chance of using the right data, increases with the amount of available data. These findings are supported by the observation that not every additional year improves the accuracy, only the overall trend is positive. This seems plausible, because especially when conditions change over time, the models also can learn behaviour that is no longer valid and which

possibly decreases future forecast performance. One should therefore not only include as much data as possible, but also carefully evaluate and also possibly shorten the training data base if necessary.

## 5 Conclusions

In this study we evaluate and compare the groundwater level forecasting accuracy of NARX, CNN and LSTM models. We examine as well sequence-to-value as sequence-to-sequence forecasting scenarios. We can conclude that in the case of seq2val

forecasts all models are able to produce satisfying results, and NARX models on average perform best, LSTMs the worst. Since CNNs are much faster in calculation speed than NARX and only slightly behind in terms of accuracy, they might be the favourable option if time is an issue. If accuracy is especially important, one should stick with NARX models models. LSTMs however, are most robust against initialisation effects, especially compared to NARX. Including past groundwater levels as

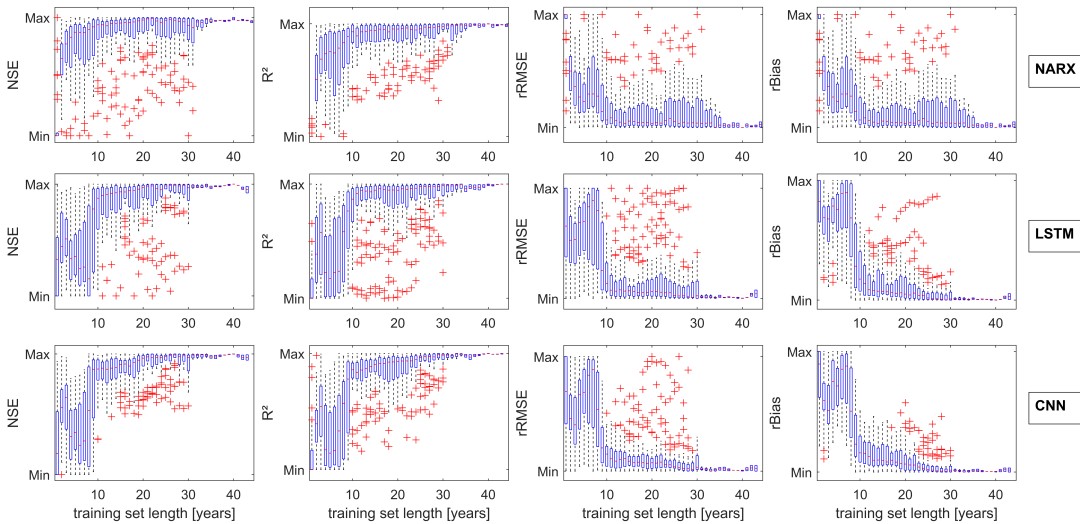

**Figure 8.** Influence of training data length on model performance.

inputs strongly improves CNN and LSTM seq2val forecast accuracy. However, all three models mostly cannot beat the naïve

model in this scenario and are therefore of no value.

Especially when no input data is available in short- and mid-term forecasting applications, sequence-to-sequence forecasting is of special interest. Again, past groundwater levels as input significantly improved CNN and LSTM performance, NARX performed almost similar in both scenarios. Overall, NARX models show the best performance (except $R^2$ values) in the vast majority of all cases. In addition to the fast calculation of NARX in this case, which almost keeps up with CNN speed, they are

clearly preferable. However, NARX models are least robust against initialisation effects, which nevertheless are easy to handle by implementing a forecasting ensemble.

We further analysed what data basis might be needed or sufficient to reach acceptable results. As expected, we found that in principle the longer the training data, the better; however, a noteworthy threshold seems to exist for about 10 years of weekly training data, below which the performance becomes significantly worse. This applies especially for LSTM and CNN models,

but was also found to probably be highly dataset specific.

The results are surprising in a way that LSTMs are widely known to perform especially well on sequential data and are therefore also more commonly applied. In this work they were outperformed by CNNs and NARX models. Additionally, Deep Learning approaches are usually preferred over traditional (shallow) neural network approaches. We showed that for this specific application (i) CNNs might be the better choice due to significantly faster calculation and mostly similar performance, and

(ii) shallow neural networks, such as NARX, should not be neglected in the selection processes. Especially NARX sequence-to-sequence forecasting seems to be promising for short- and mid-term forecasts. However, we do not want to ignore the fact





that LSTMs and CNNs might perform substantially better with a larger dataset, which better fulfills common definitions of DL-applications and where deeper networks can demonstrate their strengths, such as automated feature extraction. Since such data is usually not available in groundwater level prediction tasks yet, for the moment this remains in theory.

*Code and data availability.* All groundwater data is available for free via the web services of the local authorities (HLNUG, 2019; LUBW, 2018; MUEEF, 2018). Meteorological input data was derived from the HYRAS dataset (Frick et al., 2014; Rauthe et al., 2013), which can be obtained free of charge for non-commercial purposes on request from the German Meteorological Service (DWD). Our Python and Matlab Code is available on GitHub (Wunsch, 2020)

*Author contributions.* Andreas Wunsch: Conceptualization, Methodology, Software, Validation, Investigation, Writing – original draft prepa-
ration, Visualization. Tanja Liesch: Conceptualization, Methodology, Validation, Writing – review and editing, Supervision. Stefan Broda: Methodology, Writing – review and editing

*Competing interests.* The authors declare that they have no conflict of interest.





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
