# Peer review of "Groundwater Level Forecasting with Artificial Neural Networks: A Comparison of LSTM, CNN and NARX"

_Hydrology and Earth System Sciences, 2020_

## Referee Comment (RC1) · Anonymous Referee #1 · 23 Dec 2020

**1  General Comments**

The manuscript sets out to compare the groundwater level forecasts of three different data-driven model classes. Interestingly, they find that the simplest model (NARX) outperforms the more complicated models in almost all settings. And, it exhibits at least competitive performance over the entire testing battery.

Overall, the manuscript is honest and well written; the experiments all appear carefully executed; and the findings are both interesting and novel. I would therefore recommend accepting the contribution. I do however suggest major changes. Simply, because I

believe that the manuscript does not fully exhaust its potential. And, I do believe that with a few – but crucial – changes the paper can be improved greatly.

**2  Specific Comments**

**Framing.** The authors present their research as a comparative study between different model classes. In reality, it is a reflection upon the data-dependency of empirical modelling. In the current version of the manuscript this second version is somewhat concealed. It does not appear in the abstract and introduction, and only slowly emerges in the discussion of the manuscript. I would recommend emphasising it from the start to the end. It is closer to the underlying theme. More importantly, however, it connects the study with an important branch of environmental/empirical modelling research. Loosely speaking, the task of figuring out how much data is warranted for a given setup (for a classical reference I suggest Jakeman and Hornberger; 1993 – for a more modern flavour I would like to recommend Gauch, Mai and Lin; 2021). And, goes even further by connecting it to a common thread of machine learning literature, which attributes parts of the recent success to the availability of large-scale datasets (see for example Halevy, Norvig and Pereira; 2009). The discussion and conclusions of the current version suggests to me that the authors are aware of this "data-scarcity" theme. For some reasons they did however not commit fully to it. A reframing would not require new experiments, but would give the manuscript more clarity, while also enriching the scholarly depth of the manuscript.

**Showing**. The manuscript should contain more summary tables and explanatory depictions to guide the readers. This is not to say that information is lacking. No. A lot of information is provided throughout the manuscript! It is just distributed over the entire text and some summary tables and figures are provided in the supplement. That is already useful. However, the supplement has itself 80 pages and serves a different

purpose. I am thinking more about some kind of guide/help for readers in the style of Figure 1 (Page 6). In the following I provide some examples. They are supposed to exhibit the "form of exposition" that I am thinking about. I'd like to emphasize that they are however neither exhaustive nor imperative. My hope is that they inspire the authors to provide more clarity.

- A table with the different model setups. This would make it easier for readers to keep an overview.

- A set of graphs that shows the training/calibration development of the different models. This would allow readers to see if and how the model converged. Maybe that is something for an Appendix.

- An overview map of the basin-locations. This would provide an intuition about the scope of the study.

- A table with the different input parameters and their relation to the models. This would be a convenient look back while the results and discussion section.

- A conceptual depiction of the different models. This would be helpful to make a distinction between the models (and setups) clearer. It should probably not be super technical but illustrate the conceptual differences.

- A depiction which shows the data-availability of the different hydrographs. This would provide readers with an intuition about the actual length of the data. How many started in 1967? How many in 1994? What is their distribution?

**Naming**. I do not agree with many of the introduced terminology conventions of the manuscript. I did stumble upon some of the used conventions, and believe that other readers will too. Some adaptations could avoid this and make the manuscript more clearer. Concretely:

1. **NARX.** A nonlinear autoregressive exogenous model (NARX) can be seen as an extension of the ARIMA style time-series models (see for example Box and Jenkins; 2011) . This means that we can defined a NARX using the equation $y_t = f(y_{t-1}, \ldots, y_{t-M}, x_t, \ldots, x_{t-N}) + \epsilon_t$, where $t$ is the time-index, $y$ is the regressand and $x$ constitutes the additional exogenous variables, and $f(\cdot)$ is our nonlinear linking function. The NARX setting does not necessitate the use of a specific linking function. As a matter of fact, all of the presented approaches – the shallow network, the CNN and the LSTM – can be used as non-linear components. All approaches can be used in a normal regression setting (where past estimations or measurements of the regressand is not provided to the model); an auto-regression setting (where past estimation or measurements are provided to the model); as well the presented open-loop, and closed-loop settings. I therefore think the chosen terminology (which contrasts NARX with CNNs and LSTMs) can be difficult to understand for many readers.

2. **Coefficient of Determination.** The equation for the coefficient of determination ($R^2$, equation 2) is actually the square of Pearson's correlation coefficient (note: $r^2$). The two statistical coefficients only correspond to each other in the simple linear-regression setting (and even there only post-hoc model fit). Clearly not the setting of the study. I see several possible fixes. 1. Renaming it. 2. Renaming and decapitalizing it. Reporting Pearsons's correlation coefficient instead. Regarding point 3. I admit it is the most laborious solution and eyeballing the results I would assume that it will not change much in practise. It is however the solution I would prefer, since it is closer to general scientific practise, and has the theoretical advantage that negative correlations are exposed instead of mapped back to a positive value.

3. **Sequence-to-value.** I was not able to figure out the difference between the terms "sequence-to-value" and "sequence-to-one". If the underlying settings are the same I would recommend to use the more common terminology (to give a comparison: A search on google scholar revealed around 4,400 results for the query "sequence to one", but only 38 for "sequence to value").

Lastly, I would like to reemphasize the already existing quality of the study. I just see much potential for improvements.

**3 Technical Corrections**

I do not have many technical corrections since the paper as such is well written.

- L.32ff. I would love to have some more references to the usage of the compared models in environmental research.

- L. 121ff. I miss references/sources for the CNNs (and their development).

- Figure description of Figure 1. The text starts without capitalization and the provided description is insufficient.

- L.280. The title capitalization is used differently from the reminder of the manuscript, which seems to follow an American English style.

**4 References**

- Box, G. E., Jenkins, G. M., Reinsel, G. C. (2011). Time series analysis: forecasting and control. Vol. 734. John Wiley Sons.

- Gauch, M., Mai, J., Lin, J. (2021). The proper care and feeding of CAMELS: How limited training data affects streamflow prediction. Environmental Modelling Software, Volume 135, 2021, 104926, ISSN 1364-8152.

- Haykin, S. (1999). Neural networks: a comprehensive foundation. Prentice-Hall, Inc..

- Jakeman, A. J., Hornberger, G. M. (1993). How much complexity is warranted in a rainfall‐runoff model? Water resources research, 29(8), 2637-2649.

- Halevy, A., Norvig, P., Pereira, F. (2009). The unreasonable effectiveness of data. IEEE Intelligent Systems, 24(2), 8-12.

---

## Referee Comment (RC2) · Anonymous Referee #2 · 24 Dec 2020

The authors perform a comparison between NARX, CNN and LSTM on seq to val and seq to seq mode, over a set of wells. The authors investigate not only the performance of the models but also the computational effort required to calibrate them and the effect of the training length which are interesting and useful aspects. Another interesting and novel aspect is the combined approach to hyper parameters tuning and variable selection. The work is well written and exhaustive, making the work reproducible. The set of experiments was properly designed and explained. Therefore, I recommend only a few minor changes.

It could be interesting and useful to show a map of the study area with the wells loca-

tions and the mentioned surface water bodies.

In the introduction it could be useful to point out as a novelty aspect the approach to hyper parameters tuning and variables selection. Several other works use statistical methods to determine the input sequence length, which could have a hydrological meaning (Kisi et al. 2017; Hasda et al. 2020; Zanotti et al. 2019; Di Nunno 2020). In this case results (Tables S2 – S4) show a wide variability of the length of the input. This does not give any insight about the hydrological behaviour of the water bodies, but it could be useful in cases where the correlation is not linear.

In paragraph 2.5 could be better explained the relationship between the input delay, feedback delay and the additional GWt-1 data. Since NARX is autoregressive isn't it already considering previous groundwater levels? In table S2 you have ID GWLt-1: does it mean that you are feeding into the model more than one GWL (and the same for seq length when using GWLt-1)?

The performance of the models is well presented and discussed, but a discussion could be made also relatively to very poor performance on some wells: what could be the cause of the results of e.g. BW 781-304-2 or BW 138-019-0? Maybe it could be handy to add the length of the training set in figures S1-S68 or in their captions.

Fig. 4 and its relative discussion are in the results; it could be useful to mention in the materials and methods that you performed that analysis. Same for paragraph 4.4: it is an interesting analysis, and its methodology should be appropriately explained in the methods section and anticipated in the introduction.

Line 50-51 is not clear

Di Nunno, F., Granata, F., 2020. Groundwater level prediction in Apulia region (Southern Italy) using NARX neural network. Environ. Res. https://doi.org/10.1016/j.envres.2020.110062 Hasda, R., Rahaman, M.F., Jahan, C.S., Molla, K.I., Mazumder, Q.H., 2020. Climatic data analysis for groundwater level sim-

ulation in drought prone Barind Tract, Bangladesh: Modelling approach using artificial neural network. Groundw. Sustain. Dev. https://doi.org/10.1016/j.gsd.2020.100361 Kisi, O., Alizamir, M., Zounemat-Kermani, M., 2017. Modeling groundwater fluctuations by three different evolutionary neural network techniques using hydroclimatic data. Nat. hazards 87, 367–381. Zanotti, C., Rotiroti, M., Sterlacchini, S., Cappellini, G., Fumagalli, L., Stefania, G.A., Nannucci, M.S., Leoni, B., Bonomi, T., 2019. Choosing between linear and nonlinear models and avoiding overfitting for short and long term groundwater level forecasting in a linear system. J. Hydrol. 578, 124015.

---

## Author Comment (AC1) · 15 Jan 2021

We thank Reviewer #1 for the useful suggestions and comments. We think that these really helped to improve the quality of the manuscript. We are glad to read that our manuscript is already honest and well written and that its findings are interesting and novel. We answer each of the comments in the following in red.

We have already revised the manuscript, because the discussion is still open, we will upload the revised pdf in a few days.

**1 General Comments**

The manuscript sets out to compare the groundwater level forecasts of three different data-driven model classes. Interestingly, they find that the simplest model (NARX) outperforms the more complicated models in almost all settings. And, it exhibits at least competitive performance over the entire testing battery. Overall, the manuscript is honest and well written; the experiments all appear carefully executed; and the findings are both interesting and novel. I would therefore recommend accepting the contribution. I do however suggest major changes. Simply, because I believe that the manuscript does not fully exhaust its potential. And, I do believe that with a few – but crucial – changes the paper can be improved greatly.

Thank you.

**2 Specific Comments**

**Framing.** The authors present their research as a comparative study between different model classes. In reality, it is a reflection upon the data-dependency of empirical modelling. In the current version of the manuscript this second version is somewhat concealed. It does not appear in the abstract and introduction, and only slowly emerges in the discussion of the manuscript. I would recommend emphasizing it from the start to the end. It is closer to the underlying theme. More importantly, however, it connects the study with an important branch of environmental/empirical modelling research. Loosely speaking, the task of figuring out how much data is warranted for a given setup (for a classical reference I suggest Jakeman and Hornberger; 1993 – for a more modern flavour I would like to recommend Gauch, Mai and Lin; 2021). And, goes even further by connecting it to a common thread of machine learning literature, which attributes parts of the recent success to the availability of large-scale datasets (see for example Halevy, Norvig and Pereira; 2009). The discussion and conclusions of the current version suggests to me that the authors are aware of this "data-scarcity" theme. For some reasons they did however not commit fully to it. A reframing would not require new experiments, but would give the manuscript more clarity, while also enriching the scholarly depth of the manuscript.

Thank you for this very useful comment. We admit that this deserves definitely more attention in the manuscript. We therefore established a new section (2.6) and modified many small parts of the text to make this aspect more present in the manuscript.

See changes in Lines: 11, 16, 68-69, new Section 2.6 (lines 211ff.), 382, 414-415, 420-421,

**Showing**. The manuscript should contain more summary tables and explanatory depictions to guide the readers. This is not to say that information is lacking. No. A lot of information is provided throughout the manuscript! It is just distributed over the entire text and some summary tables and figures are provided in the supplement. That is already useful. However, the supplement has itself 80 pages and serves a different purpose. I am thinking more about some kind of guide/help for readers in the style of Figure 1 (Page 6). In the following I provide some examples. They are supposed to exhibit the "form of exposition" that I am thinking about. I'd like to emphasize that they are however neither exhaustive nor imperative. My hope is that they inspire the authors to provide more clarity.

• A table with the different model setups. This would make it easier for readers to keep an overview.

Done. We combine this suggestion with your suggestion to show conceptual depictions of the different models (Fig. 1).

• A set of graphs that shows the training/calibration development of the different models. This would allow readers to see if and how the model converged. Maybe that is something for an Appendix.

Done. We provide these figures now in the electronic supplement (Figures S72 to S141).

• An overview map of the basin-locations. This would provide an intuition about the scope of the study.

Done (Fig.3).

• A table with the different input parameters and their relation to the models. This would be a convenient look back while the results and discussion section.

We are not quite sure what you mean. Input parameters are mentioned on every graphic showing any groundwater forecast. A summary for all models can be found in the supplement, but would be overwhelming for the main part of the text.

• A conceptual depiction of the different models. This would be helpful to make a distinction between the models (and setups) clearer. It should probably not be super technical but illustrate the conceptual differences.

Done. See above (Fig. 1).

• A depiction which shows the data-availability of the different hydrographs. This would provide readers with an intuition about the actual length of the data. How many started in 1967? How many in 1994? What is their distribution?

Done (Fig.3).

**Naming**. I do not agree with many of the introduced terminology conventions of the manuscript. I did stumble upon some of the used conventions, and believe that other readers will too. Some adaptations could avoid this and make the manuscript more clearer. Concretely:

1. **NARX.** A nonlinear autoregressive exogenous model (NARX) can be seen as an extension of the ARIMA style time-series models (see for example Box and Jenkins; 2011) . This means that we can define a NARX using the equation
$y_t = f(y_{t-1}, ... , y_{t-M}, x_t, ... , x_{t-N}) + \_t$,
where t is the time-index, y is the regressand and x constitutes the additional exogenous variables, and $f(\cdot)$ is our nonlinear linking function. The NARX setting does not necessitate the use of a specific linking function. As a matter of fact, all of the presented approaches –the shallow network, the CNN and the LSTM – can be used as non-linear components. All approaches can be used in a normal regression setting (where past estimations or measurements of the regressand is not provided to the model); an auto-regression setting (where past estimation or measurements are provided to the model); as well the presented open-loop, and closed-loop settings. I therefore think the chosen terminology (which contrasts NARX with CNNs and LSTMs) can be difficult to understand for many readers.

To be clearer in contrasting the models we now describe the NARX model more detailed and also as a recurrent multi-layer perceptron. We hope this prevents misunderstanding. See beginning of (Section 2.2, Lines 89-92.)

2. **Coefficient of Determination.** The equation for the coefficient of determination ($R_2$, equation 2) is actually the square of Pearson's correlation coefficient (note: $r_2$). The two statistical coefficients only correspond to each other in the simple linear-regression setting (and even there only post-hoc model fit). Clearly not the setting of the study. I see several possible fixes. 1. Renaming it. 2. Renaming and decapitalizing it. Reporting Pearsons's correlation coefficient instead. Regarding point 3. I admit it is the most laborious solution and eyeballing the results I would assume that it will not change much in practise. It is however the solution I would prefer, since it is closer to general scientific practise, and has the theoretical advantage that negative correlations are exposed instead of mapped back to a positive value.

You are right, we corrected the naming to squared Pearson r, because indeed we calculated it this way. However, we disagree that this value does not equal the coefficient of determination because we do not compare the model inputs and outputs but the model output (simulated GWL) to the observed GWL. This of course is a linear model because we basically compare the same variables and in the best case on equals the other. The simplification of Pearson $r^2$ = $R^2$ Coeff. of. Det. is therefore true. See lines 189 and 198-199.

3. **Sequence-to-value.** I was not able to figure out the difference between the terms "sequence-to-value" and "sequence-to-one". If the underlying settings are the same I would recommend to use the more common terminology (to give a comparison: A search on google scholar revealed around 4,400 results for the query "sequence to one", but only 38 for "sequence to value").

You are right, sequence to one is the more common term in a direct comparison. Also, other terms like "one step ahead", "many to one" ore "many to many" in case of seq2seq are commonly used. As it is an important publication in this area of research, we refer to Kratzert et al. (2019), who use the term sequence to value, like we did. We therefore choose to not change to naming, however, we wanted to address your comment anyway and we added some explanations to the text. See line 62

Lastly, I would like to reemphasize the already existing quality of the study. I just see much potential for improvements.

**3 Technical Corrections**
I do not have many technical corrections since the paper as such is well written.

• L.32ff. I would love to have some more references to the usage of the compared models in environmental research.
We have added some references to recent DL models in water resources. (See Lines. 38-39.) We think that covering all environmental research would be overwhelming.
• L. 121ff. I miss references/sources for the CNNs (and their development).
Done. Line 132
• Figure description of Figure 1. The text starts without capitalization and the provided description is insufficient.
Done. Now Fig. 2, Line 178
• L.280. The title capitalization is used differently from the reminder of the manuscript, which seems to follow an American English style.
Done.

**4 References**

• Box, G. E., Jenkins, G. M., Reinsel, G. C. (2011). Time series analysis: forecasting and control. Vol. 734. John Wiley Sons.
• Gauch, M., Mai, J., Lin, J. (2021). The proper care and feeding of CAMELS: How limited training data affects streamflow prediction. Environmental Modelling Software, Volume 135, 2021, 104926, ISSN 1364-8152.
• Haykin, S. (1999). Neural networks: a comprehensive foundation. Prentice-Hall, Inc..
• Jakeman, A. J., Hornberger, G. M. (1993). How much complexity is warranted in a rainfallăARˇ runoff model? Water resources research, 29(8), 2637-2649.
• Halevy, A., Norvig, P., Pereira, F. (2009). The unreasonable effectiveness of data. IEEE Intelligent Systems, 24(2), 8-12.

Kratzert, F., Klotz, D., Shalev, G., Klambauer, G., Hochreiter, S. and Nearing, G.: Towards learning universal, regional, and local hydrological behaviors via machine learning applied to large-sample datasets, Hydrol. Earth Syst. Sci., 23(12), 5089–5110, https://doi.org/10/gghmz4, 2019.

---

## Author Comment (AC2) · 15 Jan 2021

We thank Reviewer #2 for the useful suggestions and comments. We are glad to read the overall positive judgement and we think that the suggestions really helped to further improve the manuscript. We answer each of the comments in red.

We have already revised the manuscript, because the discussion is still open, we will upload the revised pdf in a few days.

The authors perform a comparison between NARX, CNN and LSTM on seq to val and seq to seq mode, over a set of wells. The authors investigate not only the performance of the models but also the computational effort required to calibrate them and the effect of the training length which are interesting and useful aspects. Another interesting and novel aspect is the combined approach to hyper parameters tuning and variable selection. The work is well written and exhaustive, making the work reproducible. The set of experiments was properly designed and explained. Therefore, I recommend only a few minor changes.
Thank you.

It could be interesting and useful to show a map of the study area with the well's locations and the mentioned surface water bodies.
We have added a map of the study area to the text including the position of the wells and the available time series lengths for each well. (see Fig3)

In the introduction it could be useful to point out as a novelty aspect the approach to hyper parameters tuning and variables selection.
We have added a statement to the introduction. (See lines 66f.)

Several other works use statistical methods to determine the input sequence length, which could have a hydrological meaning (Kisi et al. 2017; Hasda et al. 2020; Zanotti et al. 2019; Di Nunno 2020). In this case results (Tables S2 – S4) show a wide variability of the length of the input. This does not give any insight about the hydrological behaviour of the water bodies, but it could be useful in cases where the correlation is not linear.
We are aware of using statistical methods for this purpose such as cross correlation between precipitation and groundwater levels. We even performed this ourselves in an earlier study (Wunsch et al. 2018). However, we observed that optimizing the sequence length according to the needs of the network, instead of choosing a length with hydrological meaning, can yield better results. Obtaining the best possible results was definitely the overarching goal here. In fact, one should rather try to interpret the length chosen by the optimization algorithm. However, we did not find any systematics here (for example most models even choose different sequence lengths for same wells, even same models choose different sequence lengths for different setups (seq2val / seq2seq). Additionally, we rely on a poor data basis regarding local geological information for each of the wells, which makes an analysis difficult.

In paragraph 2.5 could be better explained the relationship between the input delay, feedback delay and the additional GWt-1 data. Since NARX is autoregressive isn't it already considering previous groundwater levels? In table S2 you have ID GWLt-1: does it mean that you are feeding into the model more than one GWL (and the same for seq length when using GWLt-1)?
Yes, NARX is autoregressive, hence it feeds back the simulated groundwater level for a certain number of timesteps (defined by the size of the feedback delay). However, we can also provide the NARX simultaneously with a certain number (defined by the size of the input delay) of observed groundwater levels up to t-1 (which is the last known at time t). This way, the NARX model learns the strong relation between the past observed GWLs and the desired output (GWL(t)). This is more exact than simply feeding simulated GWLs back to the model, which have a certain error that might be large. NARX models also have a mode that is known as "open loop" or "series parallel", which follows the same idea of feeding observed values as inputs. However, in this study we do NOT replace the feedback connection as it would be the case for an open loop configuration. This applies both for seq2val and seq2seq scenarios. In the latter case of course, there is an overlap of the sequences that are simultaneously fed to the

network, however, this is the case for all parameters not only past GWLs. We included additional explanations in Sections 2.2 (instead of 2.5), because we thought the they would fit there better.
Lines 106-110., Section 2.2

The performance of the models is well presented and discussed, but a discussion could be made also relatively to very poor performance on some wells: what could be the cause of the results of e.g. BW 781-304-2 or BW 138-019-0? Maybe it could be handy to add the length of the training set in figures S1-S68 or in their captions.
You are right, this aspect deserves more discussion. We added some explanations for other wells that might be useful to understand why forecasting is more challenging and why the performance declines (Lines 303ff). We also agree that additional information on the time series length would be useful. We therefore included an additional figure (Fig.3), which visualizes the available data records for each well. We think this might be handier than giving the length for each time series in the caption.

Fig. 4 and its relative discussion are in the results; it could be useful to mention in the materials and methods that you performed that analysis.
Thank you for this suggestion, however, we kindly disagree. This was really only a side aspect and we did not systematically try to find additional input parameters to improve forecasting performance. We think that it is a nice explanation and visualization within the results section, but we do not want to make it part of the overall experimental setup described in the methods section, which would give it more weight than it deserves in the manuscript. We would feel obligated to perform a proper analysis regarding other influencing parameters, which we could not do, due to lacking data.

Same for paragraph 4.4: it is an interesting analysis, and its methodology should be appropriately explained in the methods section and anticipated in the introduction.
Thank you for this comment, indeed this got somehow lost in the previous version of our manuscript. We added an explanation on our experiments and established a new section on data dependency (Section 2.6, Lines 211ff.) Please compare also comments of Reviewer #1.

Line 50-51 is not clear
We removed the respecting sentence because it gives no added value to the manuscript anyway.

Di Nunno, F., Granata, F., 2020. Groundwater level prediction in Apulia region (Southern Italy) using NARX neural network. Environ. Res. https://doi.org/10.1016/j.envres.2020.110062

Hasda, R., Rahaman, M.F., Jahan, C.S., Molla, K.I., Mazumder, Q.H., 2020. Climatic data analysis for groundwater level simulation in drought prone Barind Tract, Bangladesh: Modelling approach using artificial neural network. Groundw. Sustain. Dev. https://doi.org/10.1016/j.gsd.2020.100361

Kisi, O., Alizamir, M., Zounemat-Kermani, M., 2017. Modeling groundwater fluctuations by three different evolutionary neural network techniques using hydroclimatic data. Nat. hazards 87, 367–381.

Zanotti, C., Rotiroti, M., Sterlacchini, S., Cappellini, G., Fumagalli, L., Stefania, G.A., Nannucci, M.S., Leoni, B., Bonomi, T., 2019. Choosing between linear and nonlinear models and avoiding overfitting for short and long term groundwater level forecasting in a linear system. J. Hydrol. 578, 124015.

Wunsch, A., Liesch, T. and Broda, S.: Forecasting groundwater levels using nonlinear autoregressive networks with exogenous input (NARX), Journal of Hydrology, 567, 743–758, https://doi.org/10/gcx5k4, 2018.